# Aggregate Online Brand Name Pharmacy Price Dynamics for the United States and Mexico

**Thomas M. Fullerton, Jr.** [1,*] and **Steven L. Fullerton** [2]

1   Department of Economics & Finance, University of Texas at El Paso, El Paso, TX 79968-0543, USA
2   Border Region Modeling Project, University of Texas at El Paso, El Paso, TX 79968-0543, USA; slfullerton@utep.edu
*   Correspondence: tomf@utep.edu; Tel.: +1-915-747-7747; Fax: +1-915-747-6282

**Abstract:** Virtual cross-border medical tourism allows many residents in the United States to purchase brand name medicines from companies in Mexico without travelling there. Monthly economic reports indicate that the online brand name pharmaceutical product prices in Mexico are noticeably lower than the corresponding internet prices in the United States. There have been very few econometric studies on how these prices are linked and the dynamic nature of those relationships. Results in this study indicate that online medicine prices in Mexico respond very rapidly to online prices changes in the high-price market.

**Keywords:** online pharmacies; brand name medicines; price dynamics; applied econometrics

**JEL Classification:** I11 Health Markets; L81 E-Commerce; M21 Business Economics

## 1. Introduction

In 2018, prescription drug expenditures in the United States exceeded USD 335 billion (CDC 2021). Due to high prices, consumers from the United States often purchase brand name pharmaceutical products in Mexico (Dalstrom et al. 2020). Those purchases are transacted both in-person and online (Fullerton and Miranda 2011; Fullerton et al. 2014). Most research related to this form of cross-border medical tourism employs point-in-time cross-sectional data sets (Dalstrom et al. 2021).

This study examines an overlooked aspect of the online brand name medicine trade. Rather than examine price differences for a single point-in-time, this effort examines whether there is a dynamic relationship between online medicine prices in the two countries. It achieves this using a data set that monitors monthly prices for the top 50 online brand name medicines sold in Mexico and the United States (Fullerton and Fullerton 2022).

## 2. Literature Review

Medication price studies tend to rely on cross sectional data (Daalen et al. 2021). Some studies have documented links between market structures and competitive effect pricing patterns (Barigozzi and Jelovac 2020; Granlund 2022). Potential time series linkages among cross-country industry aggregate prices have not received as much attention. While regulatory and other international boundary barriers may limit the strength of those pricing links, very little empirical evidence regarding that possibility has been previously conducted.

Lower priced medicines are sought by consumers due to budget constraints. As shown by the "generics paradox", there is no guarantee that the availability of lower priced drugs will cause prices to decline (Vandoros and Kanavos 2013). However, cross-sectional price data indicate that United States consumers can save more than USD 1000 annually by purchasing prescription medicines online in Mexico (Fullerton et al. 2014). There is substantial evidence that consistently confirms that brand name pharmaceuticals are substantially more expensive in the United States than in Mexico and most other countries

(Mulcahy et al. 2021), but no study of how those prices evolve over time or if cross-country movements in those prices are correlated.

Brand name medicines tend to be more expensive in lower- and middle-income countries once income and cost of living differences are taken into account, but not in dollar terms based on currency market exchange rates (Moye-Holz and Vogler 2022). Patients in the United States are able to take advantage of the latter with respect to pharmaceutical products purchased in Mexico either in-person (Fullerton and Miranda 2011) or online (Fullerton et al. 2014). This study examines whether the online prices for medicines sold to customers from both countries are correlated over time.

### 3. Data and Methodology

The University of Texas at El Paso Border Region Modeling Project collects prices every month for brand name medicines that can be purchased online in Mexico and the United States. These prices do not include value-added taxes in Mexico, sales taxes in the United States, or shipping and handling costs, nor do they include the cost of doctor appointments. International medicines can be legally imported by individual consumers in the United States as long as the amounts purchased do not exceed 90-day supplies. Furthermore, those purchases can only be for personal use (CBP 2011; Fullerton et al. 2014).

The sample period is from January 2007 through December 2021. Prices for the 50 top selling medicines are for equal dosages on a per unit basis, exclusive of shipping fees, handling charges, and taxes. Websites in both countries advertise their services in English and quote product prices in dollars. The internet sites that sell the medicines do not always offer all 50 brand name products, so multiple sites have to be sampled. Unweighted averages for both sets of prices are calculated from the data collected for each month.

Table 1 reports summary statistics for both average price variables. In line with conventional wisdom, the average price in Mexico is noticeably lower than that of the United States. However, the north-of-the-border price index is somewhat more variable than that observed for Mexico. The standard deviation of the United States online price index is double the magnitude of the southern online price index. While that is impressive, it should be noted that the difference in the magnitudes of the coefficients of variation is much more moderate. The data for both indices are slightly left-skewed, but fairly close to being symmetric. Both sets of price data are notably platykurtic.

**Table 1.** Summary Statistics.

| Variable Name | AVG_$_MEX | AVG_$_USA |
| --- | --- | --- |
| Mean | 11.28 | 18.85 |
| Median | 13.09 | 23.17 |
| Std. Dev. | 5.28 | 10.89 |
| Maximum | 18.71 | 30.51 |
| Minimum | 3.85 | 3.76 |
| Skewness | −0.341 | −0.361 |
| Kurtosis | 1.510 | 1.370 |
| CV | 0.468 | 0.578 |

Notes: Sample Period: January 2007—December 2021. Unit of Measure: United States Dollars. AVG_$_MX—Average Brand Name Medicines Price in Dollars for Mexico Internet Sites. AVG_$_USA—Average Brand Name Medicines Price in Dollars for USA Internet Sites. Std. Dev.—Standard Deviation. CV—Coefficient of Variation. Data Web Sites—goodrx.com accessed on 15 December 2021, healthwarehouse.com accessed on 15 December 2021, medicinesmexico.com accessed on 15 January 2020, medsmex.com accessed on 15 December 2018, mexmedsforyou. com accessed on 15 December 2021.

A linear transfer function autoregressive moving average (LTF ARIMA) methodology is employed to examine the dynamic relation between the two indices. For monthly time series data, the LTF ARIMA approach has proven useful in other contexts involving

international economic data for the United States and Mexico (Fullerton and Solis 2020). Due to the magnitude of the United States market for pharmaceutical products, causality between the two price indices is expected to run north-to-south. If that were not the case, then an alternative method such as vector autoregression would be required (Diebold 2007).

## 4. Empirical Results

A series of F-tests at various lags indicate weak north-to-south unidirectional causality. Given all of the regulatory barriers separating the two markets, that is not surprising. The data indicate that changes in United States brand name online drug prices are accompanied by similar changes in Mexico internet prices, but not vice versa. That implies that medicine prices in Mexico are responsive to price fluctuations in the higher income market.

A cross-correlation function is used to determine the potential lag structure governing the linkages between the two series, (Diebold 2007). The outcome indicates that the reaction of online prices in Mexico is contemporaneous with no subsequent, statistically reliable lagged responses. Parameter estimation outcomes are summarized in Table 2.

**Table 2.** Generalized Least Squares Estimation Results.

| Variable Name | Coefficient | Std. Error | t-Statistic | Probability |
|---|---|---|---|---|
| Constant | −0.0195 | 0.0358 | −0.5460 | 0.5857 |
| d(AVG_$_USA) | 0.4610 | 0.0492 | 9.3618 | 0.0000 |
| MA(4) | −0.1082 | 0.0763 | −1.4191 | 0.1576 |
| R-squared | 0.3276 | Dependent Variable Mean | 0.0475 | |
| Adj. R-squared | 0.3200 | Dep. Var. Std. Deviation | 0.6341 | |
| Pseudo R-sq. | 0.9951 | Std. Error of Regression | 0.5229 | |
| Sum Sq. Residuals | 48.1226 | Log Likelihood | −136.4433 | |
| F-statistic | 42.8730 | Prob. (F-statistic) | 0.0000 | |
| Adjusted Sample: | 2007M02–2021M12 | Included Observations: | 179 | |
| Convergence achieved after 3 iterations. | | | | |
| Coefficient covariance computed using outer product of gradients. | | | | |
| Inverted MA Roots | 0.57 | 0.00–0.57i | 0.00–0.57i | −0.57 |

The constant term in Table 2 indicates that Mexico online medicine prices tend to lose ground at a rate of roughly USD 0.02 per month. The standard error for that coefficient is fairly large, so the reliability of that estimate is not very strong. Most of the sample period corresponds to a period of relative currency market weakness for the peso, especially after 2015. That may account for some of the steady erosion of the south-of-the-border prices that occur in Table 2.

The slope coefficient indicates that every USD 1 increase (decrease) in United States online prices is accompanied by a USD 0.46 increase (decrease) in Mexico online prices for brand name pharmaceuticals. In the sample means, the magnitude of the coefficient indicates that the elasticity of online prices in Mexico with respect to those of the United States is 0.770. That estimate implies a fairly high degree of cross-border price sensitivity for brand name medicines sold online in Mexico. The computed t-statistic for this parameter estimate is 9.362 with a 0.000 *p*-value.

The overall diagnostics in Table 2 are relatively favorable. The pseudo coefficient of determination is 0.995. Given that, it is not surprising that the computed F-statistic of 42.873 has a *p*-value of 0.000. The specification does not, however, account for all systematic variation in the dependent variable. Residual serial correlation necessitates the inclusion of moving average term at lag 4. That coefficient has a somewhat large standard error, but higher log-likelihood statistic results for the equation when it is included.

As a robustness check, the sample period was shortened by seven years to cover only January 2007 through December 2014. That period is selected because it pre-dates the 2016

US presidential campaign that began in 2015, wherein several major candidates criticized trade with Mexico. Those results also indicate that the internet pharmacy prices charged in Mexico react very quickly to any north-of-the-border price variations and the slope coefficient is almost identical to that reported in Table 2. The coefficients of determination for the shorter period are larger than those reported in Table 2, potentially due to a less friendly trade environment between the two countries and due to the advent of the global pandemic in 2020 (Komkova 2019; Ceylan et al. 2020).

### 5. Conclusions

Empirical research on global pharmaceutical prices has uncovered numerous interesting commonalities and differences across international markets. This study examines dynamic aggregate price movements for brand name medicines sold over the internet in the United States and Mexico. Although brand name medicine distribution is tightly regulated in both countries, it is legal for consumers to import limited quantities for personal usage.

Over the course of the 15-year sample period, internet medicine prices in Mexico are, on average, 40 percent below the online prices charged in the United States. The prices in Mexico react very quickly to any variation in the prices in the higher-income market. Every USD 1 change in the United States average price index is matched by a USD 0.46 change in the Mexico average price index.

Based on the results reported in this exploratory effort, it seems clear that more research on this topic is warranted. This study employs simple average price measures for both economies. A logical next step would be to examine dynamic patterns among prices for these brand name medicines within a panel setting. The outcomes noted above indicate that cross-border linkages between individual online brand name pharmaceutical prices may be fairly strong.

**Author Contributions:** Conceptualization, T.M.F.J. and S.L.F.; Methodology, T.M.F.J.; Validation, T.M.F.J. and S.L.F.; Formal analysis, T.M.F.J. and S.L.F.; Investigation, T.M.F.J. and S.L.F.; Data curation, S.L.F.; Writing—original draft preparation, T.M.F.J. and S.L.F.; Writing—review and editing, T.M.F.J. and S.L.F.; Supervision, T.M.F.J. and S.L.F.; Project administration, T.M.F.J.; Funding acquisition, T.M.F.J. and S.L.F. All authors have read and agreed to the published version of the manuscript.

**Funding:** This research was funded by El Paso Water, National Science Foundation Grant DRL-1740695, Texas Department of Transportation ICC 24-0XXIA001, TFCU, UTEP Institutional Advancement, and the UTEP Center for the Study of Western Hemispheric Trade.

**Institutional Review Board Statement:** Ethical review and approval are not required for this study due to the fact that anonymous data are used that are not traceable to individuals at any time.

**Informed Consent Statement:** Not applicable.

**Data Availability Statement:** Data employed for this study are available upon request from tomf@utep.edu and slfullerton@utep.edu.

**Acknowledgments:** Helpful comments and suggestions were provided by Dan Pastor and two anonymous referees.

**Conflicts of Interest:** The authors declare no conflict of interest.

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
