# Peer review of "Aggregate Online Brand Name Pharmacy Price Dynamics for the United States and Mexico"

_economies, doi:10.3390/economies10050112_

Round 1
Reviewer 1 Report
Hi,
Great study! Table 2 is really hard to read - suggest better formatting.
In the attached pdf, I've highlighted bits of text and added comments in the right hand side - very few.
I'd be interested to extend this study to a wider range of drugs, not just the top 50 selling drugs.

Author Response
Thank you for your recommendations.
Referee 1: Corrections 1 - 3
- As recommended by Referee 1, Table 2 has been reformatted for readability.
- As recommended by Referee 1, Line 42 of Page 1 has been modified by clarifying that brand name pharmaceuticals are substantially more expensive in the United States than in Mexico and most other countries. The original version did not indicate that Mexico was one of those countries.
- As recommended by Referee 1, the capitalization error for the word “in” on Line 130 of Page 3 has been corrected.
Reviewer 2 Report
It is a well written scientific paper, my only concern is whether the relationship found in Table 2 is correlation, instead of causality. In addition, whether the correlation existing in same period or different time period.
Author Response
Thank you for your recommendations and suggestions.
Referee 2: Corrections 4 - 9
- As recommended by Referee 2, the sentence that began on Line 57 of Page 2 has been modified for grammatical purposes. The original sentence was too long and needlessly redundant. It has now been restructured and divided into two complete sentences.
- As recommended by Referee 2, a new sentence has been added beginning on Line 92 of Page 3. It clarifies that if bi-directional causality between the two markets existed, then an alternative methodology such as vector autoregression would be required for parameter estimation. A reference to Diebold (2007) is also included with that sentence.
- In response to the concerns expressed by Referee 2 with respect to estimation reliability, a second set of GLS LTF regression analysis was conducted using sample data only for January 2007 through December 2014. That sample period pre-dates the somewhat more strained trade relations that started to emerge between Mexico and the United States in 2015 as well as the global pandemic that began in 2020. A new paragraph that begins on Line 140 of Page 4 discusses those results. Because they are so similar to the full sample period results, they have not been included in the revised manuscript. That step can be taken if deemed helpful.
- In response to the concerns expressed by Referee 2 with respect to estimation and empirical results reliability, two new references have been added to the paragraph that begins on Line 140 of Page 4 .
- A bibliographic reference regarding economic effects of COVID-19 has been added to the References section beginning on Line 176 of Page 4.
- A bibliographic reference regarding strained trade relations between the United States and Mexico has been added to the References section beginning on Line 206 of Page 5.